# End-to-End Learning on
# 3D Protein Structure for Interface Prediction

**Raphael J. L. Townshend**
Stanford University
raphael@cs.stanford.edu

**Rishi Bedi**
Stanford University
rbedi@cs.stanford.edu

**Patricia A. Suriana**
Stanford University
psuriana@stanford.edu

**Ron O. Dror**
Stanford University
rondror@cs.stanford.edu

## Abstract

Despite an explosion in the number of experimentally determined, atomically detailed structures of biomolecules, many critical tasks in structural biology remain data-limited. Whether performance in such tasks can be improved by using large repositories of tangentially related structural data remains an open question. To address this question, we focused on a central problem in biology: predicting how proteins interact with one another—that is, which surfaces of one protein bind to those of another protein. We built a training dataset, the Database of Interacting Protein Structures (DIPS), that contains biases but is two orders of magnitude larger than those used previously. We found that these biases significantly degrade the performance of existing methods on gold-standard data. Hypothesizing that assumptions baked into the hand-crafted features on which these methods depend were the source of the problem, we developed the first end-to-end learning model for protein interface prediction, the Siamese Atomic Surfacelet Network (SASNet). Using only spatial coordinates and identities of atoms, SASNet outperforms state-of-the-art methods trained on gold-standard structural data, even when trained on only 3% of our new dataset. Code and data available at https://github.com/drorlab/DIPS.

## 1 Introduction

Proteins are large molecules responsible for executing almost every cellular process. Their function depends critically on their ability to bind to one another in specific ways, forming larger machines known as protein complexes. In this work we tackle the problem of paired protein interface prediction: given the separate structures of two proteins, we wish to predict which surfaces of the two proteins will come into contact upon binding. This is in contrast to the single-interface prediction problem, where one wishes to predict which parts of a single protein are likely to form interfaces. Correctly predicting protein interfaces has important applications in protein engineering and drug development.

A large number of experimental structures of protein complexes are available, but—as in many structural biology tasks—the amount of supervised data available for paired protein interface prediction remains limited. Few gold-standard cases exist in which structures are available both for two proteins bound to one another and for each of the two proteins on its own. We wondered if much larger sets of structural data might be deployed in service of tasks such as protein interface prediction.

To investigate this problem, we mine the Protein Data Bank (PDB) [1] to construct a large dataset of protein complex structures for which structures of the individual proteins on their own are not available. We introduce the Database of Interacting Protein Structures (DIPS), which comprises 42,826 binary protein interactions—an increase of more than two orders of magnitude over previously

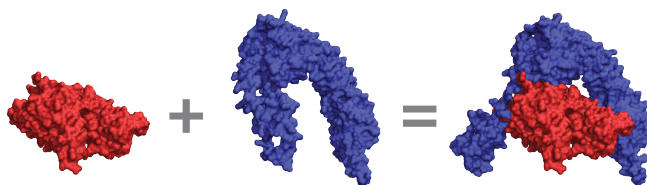

Figure 1: Protein Binding. The BNI1 protein (blue) opens up to bind to actin (red). While our method is trained only using structures of complexes such as the one at right, without any information on how the individual proteins deformed upon binding, we test on pairs of unbound structures such as those at left with minimal loss in performance.

used datasets such as Docking Benchmark 5 (DB5). However, we find that existing state-of-the-art methods are unable to effectively leverage this larger dataset, likely because the assumptions built into these methods' hand-crafted features are not robust to differences between this training data and the gold-standard test data set.

We therefore present SASNet, the first end-to-end learning method applied to interface prediction. Instead of relying on hand-engineered, high-level features, we work directly at the atomic level, using only atom positions and identities as inputs. To predict whether an amino acid on the surface of one protein interacts with an amino acid on the surface of another protein, we voxelize the local atomic environments, or "surfacelets," surrounding each of them and then apply a siamese-like three-dimensional convolutional neural network to the resulting grids. SASNet outperforms existing methods for structure-based interface prediction while leaving the door open to substantially greater performance improvements not available to competing methods, as we have so far trained on less than 3% of DIPS (due to computational limitations), whereas standard models are already using all of the gold-standard training data available to them.

There is good reason to believe that convolutional neural networks would be an appropriate fit for this problem and others in structural biology. For one, the available data is homogeneous in its underlying representation: we are given a collection of atoms $a \in \mathbb{A}$ where $\mathbb{A} = \mathbb{P} \times \mathbb{E}$ such that $\mathbb{P} = \mathbb{R}^3$ is the position space and $\mathbb{E} = \{C, N, O, S, ...\}$ is the set of possible atom element types. We are also especially interested in modeling proximal interactions due to the local nature of the underlying physical forces, a natural strength of the convolutional filters. Finally, the stacked nature of neural networks approximates the hierarchical nature of biomolecular structure: for example a protein can be progressively broken down into domains, secondary structure elements (e.g. alpha helices and beta sheets), amino acids, and finally atoms.

However, a major surprise relates to the primary source of bias in DIPS: the proteins within are provided only in their final bound form, in which their shapes almost always match perfectly with one another. This is in sharp contrast to real tests cases such as those included in DB5, in which the structures of the individual proteins typically lack shape complementarity because proteins tend to deform substantially upon binding. Even though SASNet does not explicitly account for the fact that proteins deform upon binding, when we train on DIPS and test on DB5, our method outperforms state-of-the-art techniques that exploit hand-engineered features and are trained directly on DB5.

This performance even in the face of such significant bias is especially exciting as the set of possible configurations a protein can take on when bound to a partner is a subset of all its possible configurations. Protein interfaces must take on a specific configuration upon binding in order to fit together in an energetically favorable manner (i.e., the atoms are more restricted to particular positions; see Figure 1) [2]. DIPS only contains proteins in conformations that can already fit together, whereas DB5 also contains protein conformations that require major deformations before being able to fit together. Our model's ability to perform well on DB5 indicates the model has not simply memorized the rules governing interaction in our DIPS dataset (e.g., by searching for shape complementarity). Instead, it has learned a representation that at least partially encodes the flexibility of proteins, without being explicitly trained to do so, unlike previously reported methods.

## 2   Related Work

There has been significant interest in applying machine learning methods to biomolecules such as proteins, DNA, RNA, and small drug-like molecules. Graph-based approaches have been used for deriving properties of small molecules [3], [4], [5], such as predicting the results of quantum mechanical calculations. You *et al.* [6] employed graph policy networks to generate new molecules. Another common representation for quantum mechanical calculations is based on Behler and Parrinello [7]'s symmetry functions which use manually determined Gaussian basis functions, as in [8], [9]. Gomes *et al.* [10] uses the symmetry functions for protein-ligand binding affinity prediction. Equivariant networks represent another recent and exciting line of work extending these symmetry functions [11], [12], [13]. 3D convolutional networks have been used for protein-ligand binding affinity [14], [15], [16], as well as for protein fold quality assessment [17], protein structure classication [18], fingerprint prediction [19], and filling in missing amino acids [20]. [21] use variational autoencoders to create coarse grain molecular dynamics simulations. [22] develop topology-based networks to predict biomolecular properties. These tasks differ substantially from protein interface prediction, however, in that they are much less data-limited.

Turning to the problem of paired interface prediction, methods developed by Fout *et al.* [23] and Sanchez-Garcia *et al.* [24] have the highest reported performance. They both apply machine learning techniques (graph convolutions and extreme gradient boosting, respectively) to hand-designed sequence conservation and structural features and are trained only on DB5. They choose to represent the protein at the amino acid level, and their structural features capture coarse-grained structure. These features include, for example, a measure of exposure of each amino acid to solvent and the number of other amino acids in a half-sphere oriented along an amino acid's side chain. These features do not, however, capture more detailed information such as the geometric arrangement of atoms in an amino acid's side chains. For the distinct task of single interface prediction, also known as binding site prediction, methods such as Jordan *et al.* [25], Porollo and Meller [26], Northey *et al.* [27], and Hwang *et al.* [28] also use high-level structural features to predict interfacial residues, but in a non-partner-specific manner—given a single protein, these methods predict which of its amino acids are likely to form an interface with any other protein. We choose to focus on paired interface prediction as Ahmad and Mizuguchi [29] demonstrated that partner-specific interface predictors yield much higher performance. Paired interface prediction is also of importance to protein–protein docking, the computational task of predicting the three-dimensional structure of a complex from its individual proteins. Docking software currently achieves low accuracy [30]: the lack of robust interface predictors for ranking candidate complexes has been identified as one of the primary issues preventing better performance [31].

Sequences of related proteins (e.g,. sequence conservation and coevolution) represent another source of information for addressing the interface prediction problem. The basic idea is that interfacial surfaces of a protein are typically constrained in how they can evolve, as too much variability can interrupt interactions that might be vital to protein function. For example, Ahmad and Mizuguchi [29] uses neural networks trained on such features. Given that all these interfaces are determined by the physics of actual three-dimensional interactions, the relegation of structure to a hidden and unmodeled variable leads to limitations of these approaches. The general consensus in the field is that the performance of purely sequence-based methods is approaching their limit [32]. Information about related proteins, including using known protein interactions as templates, can boost the performance of structure-based methods, but here we study the problem of how best to predict the interface between two proteins given only the structures of the two proteins — both because we wish to focus on identifying optimal structural features and because information about related proteins is not always available, particularly for designed or engineered proteins.

Our contributions to the problem of paired interface prediction include the first use of end-to-end learning, as well as learned structural features that achieve state-of-the-art performance. Furthermore, we mine the novel DIPS dataset and demonstrate that end-to-end learning instead of hand-engineering features enables us to leverage these sorts of much larger structural biology datasets—despite their inherent biases.

| Dataset | # Binary Complexes | # Amino Acid Interactions |
|---------|--------------------|---------------------------|
| DB5 | 230 | 21,091 |
| DIPS | 42,826 | 5,767,093 |

Table 1: Dataset Sizes. By training on complexes from the newly created DIPS dataset, as opposed to restricting ourselves to complexes with unbound data available such as those from DB5, we can access over two orders of magnitude more training data than would otherwise be available.

## 3 Dataset

The best existing methods for protein interface prediction rely on the Docking Benchmark 5 (DB5) dataset [30]. This gold-standard set contains most known labeled examples for the protein interface prediction problem. It is also relatively small: 230 complexes in total. Interfacial amino acids (i.e., the labels) are defined based on the structure of the two proteins bound together, but the three-dimensional structures used as input to the model are those of the two proteins when they are unbound. The data distribution therefore closely matches that which we would see when predicting interfaces for new examples, which are provided in their unbound states as we do not know the structure of the resulting complex a priori. Additionally, the range of difficulties and of interaction types in this dataset (e.g., enzyme-inhibitor, antibody-antigen) provides good coverage of typical test cases one might see in the wild. State-of-the-art methods [23], [24] further split DB5 into a training/validation set of 175 complexes, DB5-train, corresponding to DB4 (the complexes from the previous version, Docking Benchmark 4) and a test set, DB5-test, of 55 complexes (the complexes added in the update from DB4 to DB5). This time-based split simulates the ability of these methods to predict unreleased complexes, as opposed to a random split which has more training/testing cross-contamination. For comparison we also use DB5-test as our test set.

While DB5 includes only 230 complexes, the PDB contains over 160,000 structures, providing an alluring target for increasing the amount of training data available. We therefore set out to construct the Database of Interacting Protein Structures (DIPS) by mining the PDB for pairs of interacting proteins (Figure 2A). For this dataset, both the input structures to the model and the labels (that is, whether or not a given amino acid in a first protein physically contacts a given amino acid in the other protein) are derived from the structure of the complex in which the two proteins are bound together. As the PDB contains data of varying quality, we only include complexes that meet the following criteria: $\geq 500$ Å$^2$ buried surface area, solved using X-ray crystallography or cryo-electron microscopy at better than 3.5 Å resolution, only contains protein chains longer than 50 amino acids, and is the first model in a structure. As DB5 is also derived from the PDB we use sequence-based pruning to ensure that there is no cross-contamination between our train and test sets. Specifically, we exclude any complex that has any individual protein with over 30% sequence identity when aligned to any protein in DB5. This is a commonly used sequence identity threshold [33], [25], but competing methods for protein interface prediction do not employ such pruning on their training set, which may bias performance comparisons in their favor. The initial processing as well as the sequence-level exclusion yields a dataset of 42,826 binary complexes, over two orders of magnitude larger than DB5.

For both of these datasets, once these binary protein complexes are generated, we identify all interacting pairs of amino acids. A pair of amino acids — one from each protein — is determined to be interacting if any of their non-hydrogen atoms (hydrogen atoms are typically not observed in experimental structures) are within 6 Å of one another (Figure 2B) (as also used by [23], [24]). This leads to a total of over five million pairs labelled as positives in DIPS (Figure 2C, see Table 1 for exact counts). For the negatives, at train and validation we select random pairs of non-interacting amino acids spanning the same protein complexes, ensuring a fixed ratio of positives to negatives from each complex (Figure 2D, the exact ratio being defined by hyperparameter search, see Section 4). At test time we use all pairs, to match real-world conditions.

As noted previously, the distribution of structures in DIPS differs from that in DB5. For example, pre-bound proteins in DIPS have a much higher degree of shape complementarity than those in DB5, as the former exclusively comprises pairs that are in the correct conformation to bind with one another. We thus must carefully consider our model design so that we can effectively leverage this much larger set to tackle the problem of paired protein interface prediction.

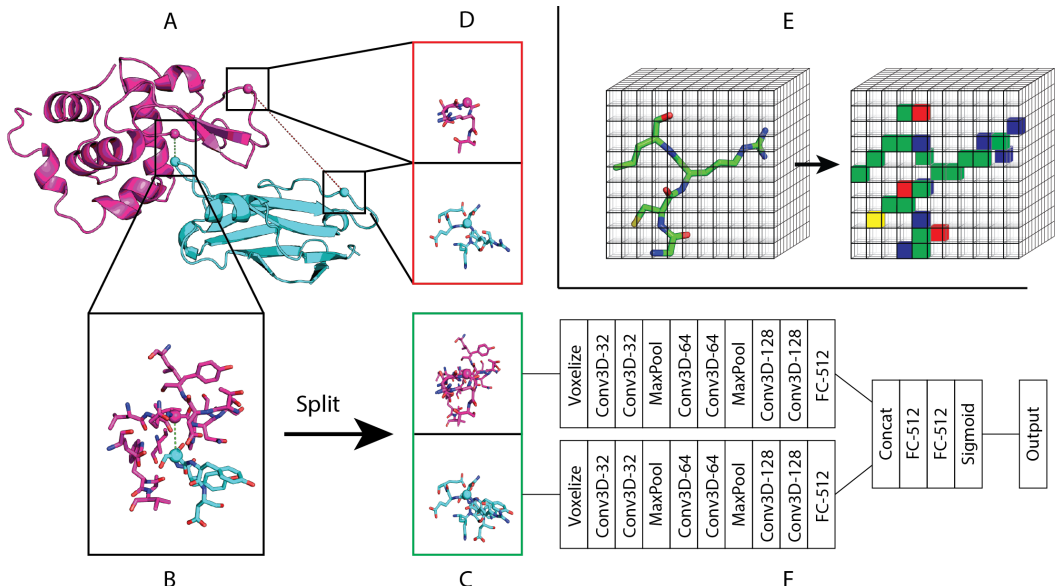

Figure 2: Protein Interface Prediction via SASNet. We predict which parts of two proteins will interact by constructing a binary classifier. To extract training examples for the problem, we start with a pair of proteins in complex sampled from DIPS (A, proteins shown in cartoon form), and from there extract all pairs of interacting amino acids (B, atoms shown in stick form). We then split these pairs to obtain our positives (C), with all remaining non-interacting pairs forming our negatives (D, negatives are down-sampled at train time, but not at test time). These pairs are then individually voxelized into 4D grids, the last dimension being the one-hot encoding of the atom's element type (E, atom channel shown as color). These pairs of voxelized representations are then fed through a 3D siamese-like CNN (F, the weights across the two arms are tied).

# 4    Method

Due to the homogeneous, local, and hierarchical structure of proteins, we selected a three-dimensional convolutional neural network as SASNet's underlying model (Figure 2F). We first focus on how to represent our pairs of amino acids and their surrounding environments in order to provide them to our network. For each amino acid in a protein, we encode all atoms of that protein within a box centered on the alpha-carbon of that amino acid — a region of 3D space that we call a "surfacelet." This encodes all structural data local to this central alpha-carbon that is provided in a given PDB structure.

To create a dense, three-dimensional, and fixed-size representation of the input, we choose to voxelize this space (Figure 2D). For each surfacelet, we lay down a grid centered on the alpha carbon of the amino acid, and record in each voxel the presence or absence of a given atom. A fourth dimension is used to encode the element type of the atom, using 4 channels for carbon, oxygen, nitrogen, and sulfur, the most commonly found atoms in protein structure (note that hydrogens are typically not resolved in experimental structures). In order to build in a notion of rotational invariance, each training example is randomly rotated, every time it is seen, across the 3 axes of rotation. At test time, we perform 20 random rotations for each example and average the predictions.

We feed the voxelized surfacelets to multiple layers of 3D convolution (Conv3D) followed by batch normalization (BN) and rectified linear units (ReLUs), and optionally layers of 3D max pooling (MaxPool). We then apply several fully connected (FC) layers followed by more BNs and ReLUs. As we are working with pairs of surfacelets, we employ two such networks with tied weights to build a latent representation of the two surfacelets, and then concatenate the results. This is a siamese-like network, but an important difference from classical siamese approaches, as introduced by Bromley *et al.* [34], arises from the nature of the task at hand. Unlike a classical siamese network, we are not attempting to compute a similarity between two objects. This can be shown by considering the nature of protein interactions: a positively charged protein surface is likely to interact with a negatively charged counterpart, even though the two could be considered very dissimilar. Instead of

| Method | CAUROC |
|---|---|
| NGF [4] | 0.843 (0.851 +/- 0.010) |
| DTNN [35] | 0.861 (0.861 +/- 0.004) |
| Node+Edge Average [23] | 0.844 (0.850 +/- 0.004) |
| Order Dependent [23] | 0.857 (0.864 +/- 0.006) |
| Node Average [23] | 0.876 (0.877 +/- 0.005) |
| BIPSPI [24] | 0.878 (0.878 +/- 0.003) |
| **SASNet** | **0.892 (0.885 +/- 0.009)** |

Table 2: DB5-test CAUROC performance. For each method we report the CAUROC of the best replicate (as selected by DIPS validation loss for SASNet, and DB5-train loss for others) as well as mean and standard deviation of CAUROC across training seeds (see section 5.1). We note that while competing methods have used all available training data, due to computational limitations our SASNet model is trained on less than 3% of our dataset, suggesting an opportunity for further performance improvements.

minimizing Euclidean distance between the two latent representation as would be done in a classical siamese network, we append a series of fully connected layers on the concatenation of the two latent representations and optimize the binary cross entropy loss with respect to the original training labels.

To determine the optimal model, we ran a large set of manual hyperparameter searches on a limited subset of the full DIPS dataset, created based on selection criteria from [36], randomly sampling a training and validation set. We varied the dataset size, number of filters, number of convolutional layers, number of dense layers, ratio of class imbalance, grid size, grid resolution, and use of max pooling, batch normalization, and dropout, and selected our models based on average performance across different training seeds on a randomly selected and held out set of DIPS. Surprisingly, most of these parameters had little effect on the overall validation performance, with the notable exception of the positive effect of increasing the overall grid size. Approximately 500 evaluation runs, each with 3 to 5 different training seeds, were computed in total.

Our model with the best validation performance involved training on 163840 examples, featurizing a grid of edge length 41 Å with voxel resolution of 1 Å (thus starting at a cube size of 41x41x41), and then applying 6 layers of convolution (each of size 3x3x3, with the 6 layers having 32, 32, 64, 64, 128, 128 convolutional filters, respectively) and 2 layers of max pooling, as shown in Figure 2F. A fully connected layer with 512 parameters lays at the top of each tower, and the outputs of both towers are concatenated and passed through two more fully connected layers with 512 parameters each, leading to the final prediction. The number of filters used in each convolutional layer is doubled every other layer to allow for an increase of the specificity of the filters as the spatial resolution decreases. We use the RMSProp optimizer with a learning rate of 0.0001. The positive-negative class imbalance was set to 1:1. The overall network is designed such that the grid feeding into the first dense layer is small enough to avoid memory issues yet large enough to capture important structural information. All models were trained across 4 Titan X GPUs using data-level parallelism, and the best model took 12 hours to train.

## 5 Experiments

To investigate the utility of the additional structural data provided in DIPS, we compare SASNet's performance to state-of-the-art methods. Furthermore, we demonstrate that competing methods trained on the larger DIPS data set see their DB5 performance severely reduced. Finally, we examine the effect of various model hyperparameters, noting that there is potential for further performance improvements via scaling to a larger fraction of the training dataset. All reported models were run across 3 to 5 training seeds.

In our performance comparisons, we utilize only information derivable from the individual protein structures provided, rather than information on evolutionarily related proteins — both because our goal is to identify the best possible structural features and because information on related proteins is not always available (see Section 2). In particular, we exclude sequence conservation and co-evolution features, and re-run the training procedures of the compared models when necessary. We note that, in

| Method | DB5 Trained | DIPS Trained |
|---|---|---|
| Node Average [23] | 0.876 (0.877 +/- 0.005) | 0.712 (0.714 +/- 0.022) |
| BIPSPI [24] | 0.878 (0.878 +/- 0.003) | 0.836 (0.836 +/- 0.001) |
| SASNet | 0.876 (0.864 +/- 0.037) | 0.892 (0.885 +/- 0.009) |

Table 3: DB5-test CAUROC for leading methods trained on DB5-train and DIPS. Competing methods with hand-engineered features experience a large drop in performance when trained on DIPS, despite its greater size. This indicates the assumptions embedded in their high-level features are not suited to the DIPS dataset. SASNet, on the other hand, increases in performance when trained on DIPS.

the real world, the interaction between two proteins is determined entirely by the structures of those two proteins, so the problem we address is a solvable one.

## 5.1 Comparison to Existing Paired Interface Prediction Methods

We start by evaluating the effectiveness of our features by comparing to top existing methods applied to interface prediction, as shown in Table 2. Graph convolutional network methods based on high-level features were pulled from the comparison in Fout *et al.* [23] and include Deep Tensor Neural Networks (DTNN) from Schütt *et al.* [35] and Neural Graph Fingerprints (NGF) from Duvenaud *et al.* [4]. Another state-of-the-art feature-engineering method is BIPSPI [24], which is based on extreme gradient boosting.

For each model, we select from available hyperparameters by choosing those with the best performance on a fixed data set, across replicates. For SASNet, this set is the validation subset of DIPS, whereas for the other methods this is DB5-train. At test time we evaluate on DB5-test, splitting the predictions by complex and computing the Area Under the Receiver Operating Characteristic (AUROC) for each one. We then calculate the median of those AUROCs. We refer to this as the median per-Complex AUROC (CAUROC). This ensures that larger complexes do not have an outsize effect on performance metrics. As our final performance metric we report the CAUROC of the replicate with the best validation performance. SASNet demonstrates superior performance without the use of any hand-engineered features, and without even directly training or validating on DB5.

## 5.2 Existing Methods Underperform with DIPS

A natural question to ask is whether SASNet's performance gains are due to the use of the larger DIPS dataset for training. If the distribution of bound and unbound were overly similar, then it would be relatively straightforward to leverage the larger size to improve performance. To investigate this, we take state-of-the-art classifiers trained on DB5 and instead train them on the same 3% of DIPS we trained SASNet on. We run this procedure on the two competing methods with the highest performing structural features, BIPSPI [24] and Node Average [23].

Instead of staying even or increasing, the performance of competing methods degrades when trained on DIPS as opposed to DB5 (Table 3). This reflects a lack of robustness to the biases inherent to DIPS. Our method, on the other hand, is robust to the use of DIPS for training, allowing us to use the larger training dataset successfully. We also observe that SASNet trained on DB5 suffers some degradation in performance due to the smaller dataset, but remains competitive with the state-of-the-art.

## 5.3 Hyperparameter Effects

Given the expense of running 3D convolutions, our best models are limited to being trained on a fraction of the full DIPS dataset. We are additionally limited by the size and resolution of the grids due to the cubic relationship between edge size and the total number of voxels. As these are problems that are surmountable through additional engineering effort and compute power, we are interested in assessing the potential benefits of scaling up along these axes. We run five training seeds per condition and plot the average and standard deviation of CAUROC across replicates.

Figure 3A shows the results of the grid size scaling tests, with resolution held fixed at 1 Å and total number of voxels allowed to vary (e.g., grid edge size of 19 would correspond to 19x19x19 voxels).

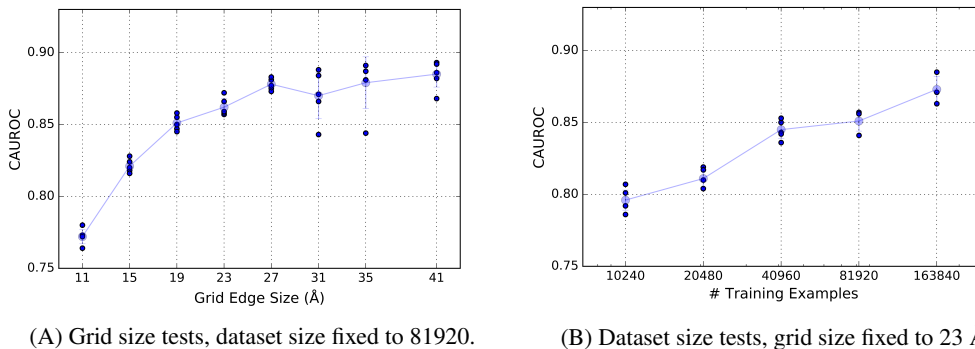

(A) Grid size tests, dataset size fixed to 81920.

(B) Dataset size tests, grid size fixed to 23 Å.

Figure 3: SASNet benefits from large input sizes (A), and has potential for being further scaled (B). We plot the DB5-test CAUROC mean and standard deviation over five different training seeds.

We notice consistent performance improvements up to a grid edge size of 27 Å, with performance increases becoming noisier and mostly tapering off afterwards. In Figure 3B, we see that larger dataset size yields consistently increasing performance, indicating that our model is capable of leveraging additional data to increase its performance and achieve state-of-the-art results.

Finally, as overfitting is always a danger with high-capacity models, we investigate even more stringent exclusion criteria, though these factors are not considered by the state-of-the-art methods to whose performance we compare. Rost [37] shows there can still be similarity between structures with sequence identities as low as 20%. Filtering out training examples with 20% or greater sequence identity to any sequence in DB5 does not significantly impact model performance, resulting in a CAUROC performance of 0.887. We also investigate structural-level pruning, removing any complexes in DIPS that share domain-domain interactions with DB5, as defined in Mosca *et al.* [38]. Such pruning does not significantly affect performance; SASNet still achieves 0.883 CAUROC.

# 6 Conclusion

In this work we introduce DIPS, a dataset for interface prediction two orders of magnitude larger than those used previously. As existing methods' hand-crafted features are unable to cope adequately with the bias present in this dataset, we create SASNet, the first end-to-end learning framework for interface prediction. We surpass current state-of-the-art results on the paired interface prediction problem while only training on proteins already in their bound configurations, without using any features identified by human experts. This is particularly intriguing as proteins are flexible structures that typically deform at multiple scales upon binding, and DIPS does not capture this deformation. The high performance on DB5 indicates our model has learned complex features beyond simple shape complementarity and has captured some notion of protein flexibility. Furthermore, the small number of assumptions made combined with the generalizability of the learned features is also of interest, as we can envision improving solutions to many data-poor structural biology problems (such as protein design and drug discovery) through training on larger, tangentially related datasets.

One hypothesis as to why SASNet's CNNs are able to generalize so well for this task is that proteins form hierarchical structures whose formation is driven primarily by local interatomic forces, making protein structures a good fit for the stacked convolutional framework. Though these properties are well understood at the lowest levels (only 22 amino acids are genetically encoded, each having a fixed atomic composition), the definitions become less precise as we move up the hierarchy. Amino acids often form secondary structure elements such as alpha-helices and beta-sheets. At a higher level, parts of the protein can form into independent and stable pieces of 3D structure known as protein domains. Many motifs are shared between proteins at all levels of this hierarchy. CNNs may be able not only to capture the known relationships between structural elements at different scales, but also to derive new relations that have not been fully characterized. Further investigation of the learned filters could yield insight into the nature of these higher-level structural patterns, allowing for a better understanding of protein structure and its relationship to protein-protein interactions.

**Acknowledgments**

The authors thank Guy Amdur, Robin Betz, Stephan Eismann, Scott Hollingsworth, Milind Jagota, Yianni Laloudakis, Naomi Latorraca, Joe Paggi, Reid Pryzant, João Rodrigues, and AJ Venkatakrishnan for their discussions and advice. This work was supported by Intel, Amazon, the National Science Foundation Graduate Research Fellowship Program under Grant No. 1147470, the U.S. Department of Energy Office of Science Graduate Student Research (SCGSR) program, and the U.S. Department of Energy, Office of Science, Office of Advanced Scientific Computing Research, Scientific Discovery through Advanced Computing (SciDAC) program.

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
