[Reviews · NeurIPS 2019]

Reviewer 1



The authors propose the first end-to-end learning model for protein interface prediction, the Siamese Atomic Surfacelet Network (SASNet). The novelty of the method is that it only uses spatial coordinates and identities of atoms as inputs, instead of relying on hand-crafted features. The authors also introduce the Dataset of Interacting Protein Structures (DIPS) which increases the amount of binary protein interactions by two orders of magnitude over previously used datasets (DB5). The results outperform state-of-the-art methods when trained on the much larger DIPS dataset and are still comparable when trained on the DB5 dataset, showing robustness when trained on bound or unbound proteins. The paper is very well written and easy to follow. The problem is well characterised and the contributions and differences with state-of-the-art methods are very clear. Originality: This work is a novel combination of well-known techniques. It combines Siamese networks with 3D convolutional neural networks and minimises the binary cross entropy loss, as opposed to minimising the Euclidean distance as in classical Siamese approaches. This work is clearly separated from previous works and the contribution is well explained. Quality: The claims in the paper are well supported by experimental analysis. There is no mathematical notation or definitions in the paper, however the architecture of the model is well described. The authors also mention that their method lacks scalability as they can only train on 3% of the data but show how the performance would keep increasing as the data size increases. Clarity: The paper is very well written and organised. It is very easy to follow and gives detailed information about the model. The problem is very clear and the contribution well defined. Significance: This work does not require hand-crafted features since it is an end-to-end framework. The method modestly outperforms competitors, but the authors show that there is a lot of room for improvement that could originate from the sheer amount of data. Other comments: • For figure 2-E the 3D representation of the surfacelet is a bit confusing, since it seems like it is just a projection of the surfacelet in 2D, but that is not the case from what I understood when reading. No depth is depicted in the second cube in the figure. A more appropriate 3D figure could improve the understanding of the representation. • I believe that the second column in Table 3 should be DB5 trained instead of DB4 trained. Numbers are the same as Table 2 where performance was calculated from training on DB5. It could be possible that the reported number is the performance on the validation set from DB4. • In section 5.1 authors define CAUROC as the median per-Complex AUROC. Is this only used for the test set? In the caption of Table 2 the mean and standard deviation is reported across training seeds, but CAUROC is the column name. • The same problem happens for Figure 3 where the label for the y-axis is CAUROC but what we see there is the Mean AUROC and standard deviation. • For figure 3 it would be nice to see the CAUROC for the test set, so that the plot reaches 0.9 as reported in the results. Also, for the grid edge size, it seems like 27 should provide a more robust choice given the very small deviation. • 194: “we use a siamese-like networks where we…” should read: we use Siamese-like networks, or we use a Siamese-like network.

Reviewer 2



1. The primary message of the manuscript - that using a much larger dataset whose properties are somewhat different than those of the testing data provides improved performance of deep 3d convolutional networks - is more interesting to researchers in the application field than to ML researchers. 2. The use of 3d convolution for protein 3d structures is not new (e.g. ref [17]) as is the overall architecture for partner specific prediction (ref [23]). 3. The fact that the model is successful despite being presented with examples that do not reflect the unbound structures, by itself does not suggest by itself that the model has learned something about protein flexibility. In related work on interface prediction using ML, other authors have observed that performance remains quite good even for "hard" examples that exhibit conformational change upon binding. This kind of statement requires some kind of justification. Minor comments: 1. The references are numbered in the Reference list but are cited by author in the manuscript itself. 2. ref [17] is published in Bioinformatics. 3. "Xue et al. (2015) demonstrated that partner-specific interface predictors yield much higher performance." That is a review paper, so they did not demonstrate that. Better to cite ref [32] for that. 4. "Due to the homogenous, local, and hierarchical structure of proteins, we selected a three-dimensional convolutional neural network as SASNet’s underlying model" From this description it sounds like you should have chosen graph convolution, which is also rotation/translation invariant. 5. "In Figure 3B, we see that the dataset size tests yield consistently increasing performance, reflecting the high degree of scalability of our model, and implying that further performance gains could be obtained with larger dataset sizes." Figure 3B does not imply what is being said here. Although it does not seem like performance has saturated, that is a possibility. Furthermore, increasing performance does not reflect scalability.

Reviewer 3



Comments: - (lines 156-158) I understand the reasoning behind removing proteins with 30% sequence identity. Would it suffice to just remove those proteins which have similarity with the DB5 test set only, and keep the ones which are similar to the DB5 training set? This way, you get a larger training dataset without having the cross-contamination issue. - Instead of the proposed voxelizing technique (which can result in abrupt jumps if the atom falls closer to the voxel boundaries), would it help to use approaches which are smoother? For example, each atom could be modeled as a sphere of a certain radius, and each voxel would get a weight based on the area overlap with the sphere. - It is clear that the competing methods' hand-crafted features are unable to use the larger dataset to improve their performance. My guess is that some features are generalizable and some are not. A more careful study of this would be very informative. - It would be nice to have some information about the voxel size, number of voxels, how many amino acids does the input typically cover, etc in the experiments section. - Would the model be able to predict contact maps, or two regions within the same protein which interact with each other? If yes, then it opens up a few more options, like getting an even bigger training dataset, pre-training the models using contact map predictors, etc. - The experimental setup of Fig 3A could be better explained. Grid size here refers to the entire window surrounding the amino acid right? Is the voxel size kept constant or scaled appropriately? What happens if it is kept constant or scaled? Originality: Novel Clarity: The presentation of the paper is clear. Significance: Moderate

[Author Response · NeurIPS 2019]

We thank the reviewers for their detailed and thoughtful feedback. We respond to each reviewer individually below.

## Reviewer 1

-Our model's computational requirements scale linearly with the size of the dataset. We have recently trained the model on much larger datasets using distributed data-level parallelism on a GPU cluster. Given a sufficient number of GPUs, it can be trained quickly on an arbitrarily large dataset.

-As suggested, we will revise the drawings in Fig. 2E to clarify that our representation is truly 3D, not a 2D projection.

-Regarding Table 3 labels: DB4 is a subset of DB5, and DB4 is the training set in Table 2 as well (hence certain numbers are the same). We will clarify this point, perhaps by dropping references to DB4 and simply referring to train/validation/test subsets of DB5.

-We only use CAUROC for the test set. This is computed per training run, and we show the mean and standard deviation of CAUROC across training seeds. So it is a mean (across training seeds) of the median (across complexes) of AUROC. We will clarify this in the Table 2 and Fig. 3 captions.

-Regarding the plot not reaching 0.9 in Figure 3: there are two factors at play here. First, we are plotting mean of CAUROC across 5 training seeds without showing the individual values. When we select the best one by validation loss, we can consistently select a higher CAUROC than the mean value. Second, our final reported results come from a dataset size of 163840, which increases performance over this plot (though we do not currently have experiments for all grid sizes at that dataset size). We will clarify this by plotting all the individual training seed CAUROCs as opposed to the standard deviation, and we will generate this plot for dataset size 163840.

## Reviewer 2

-Machine learning applied to molecular structures (in drug design, materials science, and structural biology) is a rapidly growing field of interest to machine learning practitioners—in terms of techniques, datasets, and properties of atomic data. ImageNet has had a major impact in machine learning as an effective dataset against which to pretrain for vision tasks. Likewise, our work shows how one can achieve high performance on molecular learning tasks using large sets of related data and provides a dataset for doing so.

-The use of learned features for protein interface prediction is novel. [23] feeds high-level, hand-defined features into their graph convolution architecture.

-Regarding learned flexibility: We show that unlike previous methods, our method performs well on cases involving conformational change even when the training dataset includes no conformational change at all. The training datasets for previous methods included cases that exhibit substantial conformation change. We will clarify this point.

-Regarding formatting/citations: We will make the suggested changes.

-We use a 3D CNN because graph convolutions do not directly model complex three-dimensional geometric arrangements, and proteins are complex three-dimensional structures.

-We will clarify that Fig. 3B shows that our model—unlike competing methods—is capable of leveraging additional data to increase its performance and achieve state-of-the-art results. We will avoid the term "scalability" in this context.

## Reviewer 3

-Regarding dataset availability: We have already uploaded a preliminary version of the dataset on Harvard Dataverse (the subset we trained on) and will upload the full dataset.

-Regarding pruning only against the test set and smoothing across voxels: These are good points and should only improve performance.

-Regarding analyzing why other methods fail to improve performance using the larger dataset: We agree this would be informative. For example, we can determine which high-level features don't translate well to DIPS.

-Regarding information on voxel size and number: Some of this is stated in the Methods section (voxels are 1Å in size and we use 35x35x35 voxels total per surfacelet), but we will add to and highlight this information.

-Our model could indeed be used to predict regions within the same protein that interact with one another, and we agree that this would result in significantly more training data. The key question is how to exclude parts of one surfacelet from the other so signal does not leak about each interacting component. We will point this out.

-We will better explain the experimental setup of Fig. 3A. Grid size refers to the entire window, and voxel size is kept constant. If the voxel size were allowed to vary, then the learned filters would have to encode information at different scales, which might tax the model's capacity.

-Our model scales linearly with the amount of data provided, and we have recently trained it on much larger datasets using distributed data-level parallelism on a GPU cluster. The model should indeed be useful in many molecular prediction tasks, as the reviewer notes.

[Meta-Review · NeurIPS 2019]

The main contributions of this paper, developing the SASNet model, compiling a large DIPS dataset, and empirical results, are of high significance. The paper is very clearly written and well organized, and the authors’ claims are well supported by experimental results. Although some reviewers thought that there was room for improvement in the way the idea is executed, they all agreed that the paper deserves acceptance to NeurIPS.